# Transcriptomic Profiling of Mouse Mesenchymal Stem Cells Exposed to Metal-Based Nanoparticles

**DOI:** 10.3390/ijms26157583

**Published:** 2025-08-05

**Authors:** Michal Sima, Helena Libalova, Zuzana Simova, Barbora Echalar, Katerina Palacka, Tereza Cervena, Jiri Klema, Zdenek Krejcik, Vladimir Holan, Pavel Rossner

**Affiliations:** 1Department of Toxicology and Molecular Epidemiology, Institute of Experimental Medicine of the Czech Academy of Sciences, 14220 Prague, Czech Republic; 2Department of Computer Science, Czech Technical University in Prague, 16000 Prague, Czech Republic

**Keywords:** mouse mesenchymal stem cells, nanoparticles, in vitro exposure, whole-genome expression analysis of mRNA and miRNA

## Abstract

Mesenchymal stem cells (MSCs), i.e., adult stem cells with immunomodulatory and secretory properties, contribute to tissue growth and regeneration, including healing processes. Some metal nanoparticles (NPs) are known to exhibit antimicrobial activity and may further potentiate tissue healing. We studied the effect of Ag, CuO, and ZnO NPs after in vitro exposure of mouse MSCs at the transcriptional level in order to reveal the potential toxicity as well as modulation of other processes that may modify the activity of MSCs. mRNA–miRNA interactions were further investigated to explore the epigenetic regulation of gene expression. All the tested NPs mediated immunomodulatory effects on MSCs, generation of extracellular vesicles, inhibition of osteogenesis, and enhancement of adipogenesis. Ag NPs exhibited the most pronounced response; they impacted the expression of the highest number of mRNAs, including those encoding interferon-γ-stimulated genes and genes involved in drug metabolism/cytochrome P450 activity, suggesting a response to the potential toxicity of Ag NPs (oxidative stress). Highly interacting MiR-126 was upregulated by all NPs, while downregulation of MiR-92a was observed after the ZnO NP treatment only, and both effects might be associated with the improvement of MSCs’ healing potency. Overall, our results demonstrate positive effects of NPs on MSCs, although increased oxidative stress caused by Ag NPs may limit the therapeutical potential of the combined MSC+NP treatment.

## 1. Introduction

Nanoparticles (NPs), i.e., particles with at least one dimension less than 100 nm, are, due to their unique properties, applied in many areas of human life. Among other uses, such as industrial use in electronics, materials, textiles, and energy, they are widely utilized in medical applications. Due to the antimicrobial potential of some of these substances, they are used for medical instrument coating, serve as diagnostic markers and drug delivery vehicles, and as additions for the acceleration of wound healing [1,2,3]. However, negative effects (associated mainly with the solubility and toxicity of the parent substance) of exposure to NPs have been described. These results have reported direct interaction with DNA or increased production of reactive oxygen species (ROS), which led to the disruption of various cell processes [4,5]. For example, titanium, nickel, platinum, zinc, carbon, silica, and silver NPs changed the expression profiles of numerous genes, activated the pathways of inflammatory responses, altered the cell cycle, or induced oxidative stress [6,7,8,9,10,11,12]. Due to these mixed impacts, the use of NPs in medical applications should be studied further.

Several metal-based NPs, including Ag, CuO, and ZnO NPs, have attracted attention as potential candidates for new therapies since they exhibit well-known antimicrobial properties [13]. Ag NPs are commonly used in medical practice; they are active against multiple Gram-positive and Gram-negative bacteria, and the proposed mechanisms of action involve disruption of bacterial membranes, interaction with sulfur and phosphorous groups presented in DNA and proteins, and release of toxic ions that interact with cellular components and alter cellular components and metabolic pathways [14]. Similarly to Ag, ZnO NPs also cause membrane damage to Gram-positive and Gram-negative bacteria. Due to the high electrical conductivity, they have a strong oxidizing character that can destabilize the cytoplasmic membrane and induce oxidative stress that leads to the inhibition of protein synthesis and DNA replication. ZnO NPs can also cause damage by Zn^2+^ ions released from the dissolution of ZnO in aqueous solution, disrupt the cytoplasmic membrane, and act as an inhibitor of the glycolytic enzyme through thiol group oxidation due to specific affinity for the sulfur group [15]. The primary mechanisms by which CuO NPs exert antibacterial activity are adsorption onto the surface of bacterial cells due the electrostatic forces, thus increasing cell permeability and facilitating internalization; generation of ROS through Fenton-type/Haber–Weiss reactions and also through their photocatalytic activity; and release of Cu^2+^ ions that bind thiol and phosphorous groups and destruct proteins and DNA. The sensitivity of Gram-positive bacteria to CuO NPs is higher than that of Gram-negative bacteria [16]. Unlike Ag NPs, ZnO and CuO are not routinely used in medical practice. ZnO NPs have recently become one of the most prominent metal oxide NPs in nanomedicine and are extensively tested for potential applications as antibiotics, antioxidants, anti-diabetics, cytotoxic agents, or antiviral treatment [17]. The utilization of CuO NPs is expected in nanomedicine as antibacterial and antifungal agents suitable for wound healing, antiparasitic agents, anticancer treatments, and others. Their high toxicity in vertebrates and invertebrates is, however, the limiting factor [18]. Other studies have demonstrated application of NPs as antioxidant agents, such as coatings containing TiO_2_ for treatment of rheumatoid arthritis [19] or a complex of allomelanin NPs with ruthenium NPs with a combined activity of oxygen generation and ROS removal suitable for treatment of diabetic foot ulcers [20].

Mesenchymal stem cells (MSCs) represent a unique population of adult stem cells with high regenerative potential. They contribute to tissue growth and regeneration due to their immunomodulatory and secretory properties (the production of growth factors and cytokines) [21,22,23]. Overall, adult stem cells are found in small populations in nearly all tissues of an adult organism and can differentiate into various cell types of other tissues or organs. When needed, MSCs are mostly isolated from adipose tissue or bone marrow. Treatment with MSCs supports re-epithelization and wound healing and enhances angiogenesis [24], but this positive effect could be affected by bacterial infection. Since antibiotic resistance often complicates the fight against bacteria, the simultaneous use of NPs with antimicrobial properties and MSCs represents a promising strategy to effectively eliminate bacterial infection and improve wound healing.

As suggested previously, the use of NPs may have both a positive and a negative impact on stem cells or healing processes. For example, Ag NPs may impair stem cell proliferation or differentiation [25] and cause DNA damage at sub-cytotoxic concentrations [26]. Besides these unfavorable effects, it has also been described that Ag NPs promote the osteogenesis and chondrogenesis of MSCs, thus improving, e.g., bone fracture healing and urothelial tissue regeneration [27,28,29]. Toxicity of ZnO NPs towards MSCs including oxidative stress and DNA damage has been reported [30,31]; however, enhanced osteogenesis as a positive effect was also observed [32,33]. Increased genotoxicity and ROS production was measured in MSCs following exposure to CuO NPs, while differentiation remained unaffected [34].

In our previous works, we focused on the interaction of NPs with well-known antimicrobial properties (Ag, CuO, and ZnO NPs) and mouse MSCs and investigated the various cellular and molecular impacts. MSCs were exposed to these particles for various times and concentrations that were selected to be non-cytotoxic. It was shown that the treatment exerted negative effects on the diverse cell functions, such as metabolic activity, differentiation potential, production of cytokines, and the expression of genes associated with immunoregulatory molecules or phenotypic markers [23,35]. Furthermore, the induction of ROS production, lipid peroxidation, and DNA fragmentation and oxidation occurred after the exposure as well as the alteration of the cell cycle and an increase in sensitivity to apoptosis in specific exposure conditions [36]. All these results suggest that the therapeutic potential of MSCs might be negatively impacted by the presence of NPs.

The current study is a follow-up of our previous work that investigated the impact of the interaction of NPs with MSCs on selected toxicity endpoints. Here we performed a whole genome transcriptomic analysis to obtain a comprehensive insight into the molecular mechanism of MSC–NP interactions at the transcriptional level. We aimed to reveal specific gene expression patterns reflecting the unique mode of action of each individual type of NPs in MSCs. We further aimed to detect the miRNA expression profiles as these molecules are key regulators of gene expression; they are known to play an important role in many physiological and pathological processes and are considered to be novel highly specific biomarkers of exposure. In this regard, we identified common mRNAs and miRNAs that were differentially expressed in response to all NPs as well as those which were unique for each individual NP types. We further employed pathway enrichment analysis to find common and specific deregulated processes and pathways. We also searched for significantly correlated mRNA–miRNA pairs in order to explore the possible involvement of epigenetic regulatory mechanisms.

## 2. Results

In this section, we summarize mRNA/miRNA expression alteration following the exposure of MSCs to three doses of nanomaterials (Ag, CuO, and ZnO NPs) when compared to non-exposed controls, as well as the possible influence of the MSC–NP interactions on the relevant biological pathways. The differentially expressed genes (DEGs) had to meet the criteria of adjusted *p*-value < 0.05 (statistical significance) and log_2_FC > 0.58 or <−0.58 (biological relevance).

### 2.1. mRNA Analysis

In total, in seven out of nine comparisons, we detected more than 100 DEGs (Table 1). The highest number of DEGs was found after the exposure of MSCs to the high dose of Ag NP (230) followed by the low dose of the same nanomaterial (216). An overview of all DEGs in each condition is given in Appendix A.

The analysis of common/unique DEGs showed that the exposure to Ag NPs caused deregulation of the highest number of individual genes (393), almost twice as many as the exposure to CuO and ZnO NPs. The gene deregulation effect was exposure-dose-dependent. Each exposure dose resulted in the differential expression of various genes, while only a relatively small number of genes were differentially expressed in more than one condition, with the lowest number of common DEGs after exposure to all three doses (Figure 1).

For the purposes of this study, the most biologically relevant genes were the ones which were differentially expressed regardless of the exposure dose—their expression was altered after all three doses of the nanomaterial. In the case of Ag NPs, 51 genes were commonly differentially expressed, while after exposure to CuO NPs, 36 common genes were detected. The lowest number (6) was found after exposure to all three doses of ZnO NPs (Figure 1). Out of these common DEGs, the following table summarizes the top five down- and upregulated genes based on the expression changes after the exposure to the high dose of the corresponding nanomaterial (Table 2).

Strong deregulation of several genes was shared among more than one nanomaterial. For example, *Sp7* (transcription factor, plays a major role in driving the differentiation of mesenchymal precursor cells into osteoblasts) as well as *Alpl* (alkaline phosphatase, involved in mineralization of developing bones and teeth) were found downregulated in all comparisons (in the case of *Alpl* and CuO NPs, the transcript was not in the top five list indicated in Table 2). Upregulation of *Mt1* and *Mt2* (metallothioneins, responsible for protection against metallic ion toxicity and oxidative damage, maintain the homeostasis of metallic ions) was detected in eight out of nine comparisons. These genes were not differentially expressed after exposure to the low dose of ZnO NPs, but on the contrary, they were the genes with the most increased expression after the exposure to the high dose of this nanomaterial. A complete list of genes with commonly altered expression is indicated in Appendix A.

### 2.2. mRNA Affected Pathways

Based on the lists of DEGs, a functional enrichment analysis was performed to discover the affected pathways. The detected number of biological pathways is related to the number of genes which were differentially expressed after each condition. Therefore, the highest number (27) of detected pathways was discovered after the exposure to the high dose of Ag NPs, while there was no affected pathway after the exposure to the low dose of ZnO NPs (Figure 2).

Similar to DEGs, the effect on most of these pathways was dose-dependent. Out of 48 unique pathways (Appendix A), only ten were detected in more than one condition (Figure 2); the remaining 38 pathways were dose- and nanomaterial-specific. The Neutrophil degranulation and Antiviral mechanism by IFN-stimulated gene pathways were found to be altered after exposure to all doses of Ag NPs. The following pathways were affected after exposure in three conditions combining two NPs: Innate Immune System (low and medium dose of Ag and high dose of CuO NPs) and Post-translational protein phosphorylation and Regulation of Insulin-like Growth Factor (IGF) transport and uptake by Insulin-like Growth Factor Binding Proteins (IGFBPs) (low and medium dose of CuO and medium dose of ZnO NPs) (Figure 2). Even though the detection of specific pathways was dependent on the exposure conditions, many of these pathways are related to important actions in the body, e.g., reaction of the immune system (Appendix A).

A list of the three most significantly affected pathways for each exposure condition is presented in Table 3. Due to more mRNAs being upregulated after the exposure to Ag NPs, we can assume that most of the affected pathways were induced. A close functional connection of two significant pathways (Post-translational protein phosphorylation and Regulation of Insulin-like Growth Factor (IGF) transport and uptake by Insulin-like Growth Factor Binding Proteins (IGFBPs)) is suggested because the same DEGs are involved in both of them after three different exposures (low and medium dose of CuO and medium dose of ZnO NPs).

### 2.3. miRNA

In comparison to mRNA expression changes after exposure of MSCs to various doses of three nanomaterials, fewer results were observed for differentially expressed miRNAs (DEmiRNAs). Only in two conditions (high dose of Ag and low dose of CuO NPs) were more than 10 miRNAs differentially expressed with a slightly higher number of upregulations in both exposures (Table 4, the complete list of DEmiRNAs is shown in Appendix A).

Although a low number of DEmiRNAs were detected, unlike the mRNA, their expression changes were shared among more conditions. For example, two DEmiRNAs from the mmu-miR-126 family were observed in almost all conditions. Ten common DEmiRNA were also found after exposure to a high dose of Ag and a low dose of CuO NPs (Figure 3).

### 2.4. miRNA-Affected Pathways

Due to the low number of DEmiRNAs in most of the exposure conditions, the pathway analysis was performed only for exposure to high dose of Ag and low dose of CuO NPs to ensure biological relevance. After exposure to a high dose of Ag NPs, 61 pathways were altered (due to the deregulation of 46 miRNAs). In the case of a low dose of CuO, 29 pathways were influenced (due to deregulation of 16 miRNAs). Twenty of these pathways were shared between both exposure conditions (Appendix A).

The top five most significant pathways for these conditions are listed in Table 5. Between the most significantly affected pathways after exposure to Ag and CuO NPs, some of the pathways were shared (proteoglycans in cancer and the thyroid hormone signaling pathway), which suggests a similar exposure effect on the studied system. Overall, a high number of pathways are related to various cancer processes, or in the case of the high dose of Ag NPs, to fatty acid synthesis (Appendix A).

### 2.5. miRNA–mRNA Interactions

The final analysis which was performed focused on the miRNA–mRNA interactions. For this output, the individual exposure doses for each NP were grouped together to make the analysis more robust and to generally define the interaction of miRNA and mRNA after the MSC exposure to NPs (regardless of the dose). For each NP, a list of significant interactions (originating from the lists of DEmiRNAs and DEGs) was calculated (Appendix A). The highest number of significant interactions was observed after the exposure to ZnO NPs (110); on the contrary, the lowest number of was detected for CuO (43). After the exposure to Ag NPs, 70 significant miRNA–mRNA interactions were discovered.

For the top 50 significant interactions (in the case of CuO for all 43), interaction plots showing the relations were constructed (Figure 4). In these top interactions, most of the miRNAs were found in Ag NPs (5), followed by ZnO (4) and CuO (3). Interestingly, the interactions of two miRNAs from the mmu-miR-126a family were detected after the exposure to all three NPs, and in the case of Ag and ZnO NPs, these two miRNAs reached the highest number of interactions. The exposure to ZnO impacted 42 interactions of mmu-miR-92a-1-5p (Table 6).

## 3. Discussion

Metal-based NPs are among the most frequently used nanoproducts in biomedical applications due to their antimicrobial effect at the nanoscale level. To enhance wound healing and improve tissue regeneration, the simultaneous application of antimicrobial metal NPs and MSCs has been recently tested. However, their toxicity may limit their use for medical purposes. In our current study, we applied a transcriptomic approach to reveal the global gene expression changes and the modulated biological processes and pathways in response to the activity of Ag, CuO, and ZnO nanoparticles in mouse MSCs. We further focused on the differential expression of miRNAs as these molecules mediate an important post-transcriptional regulation of gene expression. Below, we discuss the key modulated processes together with the contributing DEGs and the involvement of DEmiRNAs.

### 3.1. Detoxification of Metal Ions

In our study, a strong induction of *Mt1* and *Mt2*, genes encoding metallothioneins (MTs), was observed following the exposure to all NPs. Metal ion release is an important factor related to the toxicity of metal-based NPs [37]. Ag^+^ ions released from Ag NPs are biologically active and can mediate the antimicrobial effect as well as leading to significant cytotoxicity in mammalian cells [38]. Metal ions released from CuO and ZnO NPs were also identified as one of the major factor driving their toxicity in various organisms [39,40]. MTs are proteins that bind to heavy metals and play an essential role in protection against oxidative damage, maintenance of heavy metal homeostasis, and detoxification. They exhibit excellent antioxidant activity and effectively scavenge free radicals and mitigate oxidative stress damage. MTs display a neuroprotective and anticancer effect and reduce inflammation [41]. The study of Balfourier et al. (2022) [42], which is based on a meta-analysis of the publicly available transcriptomic data, showed that several metal-based nanoparticles, including those containing Zn, Cu, and Ag, trigger a common cell response governed by MTs or MT-related genes, which are implicated in Zn and Cu homeostasis, heavy metal detoxification, and cellular redox chemistry.

### 3.2. Neutrophil Degranulation, Immune System, Innate Immune System, and Antiviral Mechanism by IFN-Stimulated Genes

MSCs have been reported to exhibit a modulatory effect on the innate immune system [23]. An important factor in tissue repair and immune modulation mediated by MSCs is the production of extracellular vesicles (EVs). EVs represent a tool for cell–cell communication and delivery of molecules. They carry various molecules such as nucleic acids (DNA, mRNAs, and microRNAs) and proteins (cytokines, chemokines, growth factors, interleukins, and transcription factors) in order to act as paracrine/endocrine effectors, enhance the reparative process, and promote the relevant anti-inflammatory/resolutive actions in the target tissue [43]. The pathways of neutrophil degranulation and immune system pathways which were deregulated in response to all doses of Ag NPs and the high dose of CuO NPs indicate the generation of EVs. Similarly, in another relevant study, proteomic analysis of MSC-derived EVs from rats under a normoxic or hypoxic condition revealed the modulation of protein expression involved in the same pathways (“Neutrophil degranulation”, “Immune system”, and “Innate immune system”), among others [44]. Several proteins encoded by genes upregulated by Ag NPs in our study (*Aldoc, Hspa1a,* and *Hspa1b*) were confirmed as components of MSCs-EVs [44,45]. Other proteins were detected as part of EVs of various origin, e.g., metalloproteinase MMP9, which was identically upregulated by Ag and CuO NPs, is frequently detected in EVs [46]. Importantly, EVs derived from MSCs are considered as a promising alternative tool to cell therapy in regenerative medicine as they possess several benefits in comparison to the parental cells [47].

Interferon-γ (IFN-γ) can regulate the immunomodulatory function of MSCs. Deregulation of the “Antiviral mechanism by IFN-stimulated genes” pathway induced by the low dose of Ag NPs may suggest a transcriptional response to IFN-γ and production of IFN-γ-inducible genes. A study by Ren et al. (2008) demonstrated that IFN-γ elicits immunosuppression and thus enhances the therapeutic effect of MSCs [48]. On the other hand, in Holan et al. (2023) [49], it was demonstrated that the tested NPs have a negative impact on the production of various cytokines and growth factors that are essential for healing and tissue regeneration.

### 3.3. HDL Remodeling

MSCs have the potential to differentiate into adipogenic, osteogenic, or chondrogenic cells. Under specific conditions, which are determined by microenvironmental factors such as cytokines, hormones, and growth factors, a certain differentiation process is preferred. An impairment of the osteogenesis–adipogenesis balance is associated with the onset and progression of several human diseases, such as obesity, osteosclerosis, and osteoporosis. The modulated pathway “HDL remodeling” together with the upregulated genes involved in this pathway (*Apoe, Ptlp,* and *Abcg1*) may indicate that adipogenesis was enhanced specifically upon exposure to a low dose of Ag NPs. The role of APOE in lipid accumulation and adipogenic differentiation has been suggested [50]. ABCG1 is a cholesterol and phospholipid transporter and also regulates adipogenesis and fat accumulation. In another relevant study, the authors showed that downregulation of the *Abcg1* gene promoted osteogenesis, indicating a possible role of ABCG1 in the switch of adipogenesis/osteogenesis [51]. The expression of *Ptlp* has been associated with the adipogenic differentiation of human multipotent stem cells [52]. Lipid remodeling has been shown to occur during adipogenesis. The effects of Ag, CuO, and ZnO NPs on the metabolic and functional properties of MSCs was investigated in our recent study [35]. Besides the negative effects exerted by all NPs on several tested parameters such as the expression of phenotypic markers, metabolic activity, differentiation potential, the expression of genes for immunoregulatory molecules, and the production of cytokines and growth factors, we detected an impact on the differentiation potential of MSCs. The results showed that adipogenesis was inhibited by Ag and CuO NPs but not by ZnO. Alternatively, osteogenesis was enhanced in the presence of all NPs. In accordance with this study, Zhang et al. (2015) demonstrated a beneficial effect of Ag NPs on bone fracture healing; Ag NPs promoted mouse MSC proliferation and osteogenic differentiation in vitro [27]. Similarly, in He et al. (2020), enhanced osteogenesis during bone reconstruction after the application of Ag NPs to human MSCs was shown [53]. In another study by He et al. (2015), the authors observed a positive effect of Ag NPs on chondrogenesis [28], while in Sengstock et al. (2014), the authors reported rather the opposite effect: an inhibition of adipogenic and osteogenic differentiation was found following exposure to a subtoxic dose of Ag NPs [25]. In the study of He et al. (2016), the authors described enhanced adipogenesis compared to osteogenesis in human MSCs through oxidative stress generated by Ag-coated NPs [54]. No effect on osteogenic differentiation in human MSCs after the application of Ag NPs was observed in the study by Liu et al. (2014) [55]. Our current results however indicate that Ag NPs may rather enhance adipogenesis while osteogenesis is repressed. This is further strongly supported by the fact that *Sp7*, the gene coding the transcription factor which plays a critical role in the activation of osteogenesis, and *Alpl*, a well-documented marker of osteogenesis, were among the top significantly downregulated individual genes following the exposure to all NPs. Recently, mature adipocytes and their progenitors have emerged as critical regulators of tissue regeneration, prevention of fibrosis, and tissue damage [56]. Collectively, the effects of Ag NPs on differentiation are complex and depend on various factors. While some studies suggest that Ag NPs can promote bone formation and fracture healing, others indicate potential cytotoxicity and inhibitory effects on osteogenesis. There is also no clear conclusion concerning the effects of Ag NPs on adipogenic and chondrogenic differentiation of MSCs. The contradictory results reported in the aforementioned studies might be attributed to the variability in the physico-chemical properties of the used nanomaterial such as concentration, surface coating, or size [57,58]; therefore, optimal NP exposure conditions are required to obtain appropriate differentiation of MSCs. It should also be considered that gene expression profiling serves as a predictive assay to reveal the potentially affected processes, and its results should be confirmed by functional assays.

### 3.4. Calcium Regulation in Cardiac Cells, Calcium Signaling Pathway, cGMP–PKG Signaling Pathway, and Relaxin Signaling Pathway

Ca^2+^ is critical for stem proliferation, its differentiation, and maintaining stem cell potential. We found the *Ednrb* gene for endothelin 1-receptor B upregulated by CuO and ZnO NPs, while the *Adcy1* gene coding adenylate cyclase 1 was downregulated in response to all NPs. EDNRB is involved in the induction of adipogenic differentiation of adipose-derived MSCs [59]. Adenylate cyclase induces cAMP synthesis and activates cAMP signaling, a process which has been demonstrated to favor osteogenesis to adipogenesis [60]. Upregulation of *Rgs2* and downregulation of *Rgs4* was observed after the Ag NP treatment. Similar inverse regulation of the expression of *Rgs2* and *Rgs4* (*Rgs2* upregulated and *Rgs4* downregulated) during adipogenesis was found in human MSCs [61]. Overall, our data suggest that all NPs regulate Ca^2+^ signaling and the associated processes such as cAMP/cGMP signaling, possibly leading to promoted adipogenesis.

### 3.5. Extracellular Matrix Organization, ECM–Receptor Interaction

The extracellular matrix (ECM) is a crucial component of the microenvironment that surrounds MSCs and contributes to the cell survival and self-renewal/differentiation balance [62]. A disrupted balance between synthesis and the breakdown of ECM constituents may lead to various pathological situations. We identified several ECM-related genes to be upregulated following exposure to medium and high doses of ZnO and Ag NPs. A downregulation of *Spp1* was observed. *Spp1* is expressed in mesenchymal cell differentiation and is related to cell migration and osteogenesis. Interaction of SPP1 with integrins, ECM cell surface receptors, regulates cell adhesion, survival, migration, and immune response and is critical for the lineage determination of MSCs. A blockage of SPP1 function resulted in promotion of adipogenesis and the simultaneous inhibition of osteogenesis [63]. Similarly, in another relevant study, it was demonstrated that loss of *Spp1* suppressed proliferation, osteogenic differentiation, mineralization, and angiogenic potential of MSCs [64]. Tnn, another downregulated gene, belongs to the group of matricellular proteins and acts as an adhesion modulatory protein. It can influence cell adhesion, migration, and differentiation. Tnn is primarily expressed in sites of osteogenesis [65]. We observed several upregulated genes such *Fbn2*, *Eln,* and *Thbs4* that are involved in wound healing. Fibrillins are engaged in the formation of microfibrils, components of elastic fibers, and are found in the ECM of many tissue types including mesenchyme-derived connective tissues. Fibrillin (*Fbn2*) and tropoelastin (*Eln*), another elastic fiber protein, possibly play an important role in wound healing [66]. THBS4 is a key regulator of tissue growth and remodeling and is associated with tissue regeneration and numerous pathological processes that are characterized by increased proliferation and migration. It was demonstrated that THBS4 substantially contributed to the healing of skin wounds in vivo and in vitro in humans as well as in mice [67].

These results further confirm the potency of NPs to suppress osteogenesis and enhance adipogenesis and suggest the important role of NPs in the activation of ECM-related proteins that may enhance tissue regeneration.

### 3.6. Drug Metabolism-Cytochrome P450, Glutathione Conjugation, Fluid Shear Stress, and Atherosclerosis

The deregulation of the pathways “Drug metabolism–cytochrome P450”, “fluid shear stress and atherosclerosis”, and “metabolism of xenobiotics by cytochrome P450” and induction of GST enzymes was detected in MSCs following the incubation of MSCs with a high concentration of Ag NPs. The exposure to Ag NPs has been associated in many studies with oxidative stress due to the generation of free radicals such as superoxide anions, hydrogen peroxide, and hydroxyl radicals. Glutathione-S transferases are key enzymes that participate in scavenging and detoxification of ROS generated by Ag NPs [68]. Some studies have reported that ROS are involved in the regulation of stem cell differentiation. An increase in ROS generation was observed during adipogenesis in rat and human MSCs [69,70]. In a study by He et al. (2016), the authors demonstrated that Ag ions released from silver-coated nanoparticles activated ROS and possibly enhanced the adipogenic capacity of human MSCs [54]. In our recent study [36], multiple toxicity endpoints in mouse MSCs were tested in response to the application of Ag, CuO, and ZnO NPs. Numerous adverse changes such as generation of ROS, DNA damage, increased sensitivity of MSCs to apoptosis, and an altered cell cycle were found. In the current study, enhanced expression of GST-1 enzymes and other genes involved in “drug metabolism–cytochrome P450” confirmed that Ag NPs elicit the genotoxic and oxidative stress response, similarly to what has been reported previously [36]. In contrast, this response on the gene expression level was not observed for CuO and ZnO NPs.

### 3.7. Post-Translational Protein Phosphorylation, Regulation of Insulin-like Growth Factor (IGF) Transport and Uptake by Insulin-like Growth Factor Binding Proteins (IGFBPs)

Deregulation of the “Post-translational protein phosphorylation“ and “Regulation of Insulin-like Growth Factor (IGF) transport and uptake by Insulin-like Growth Factor Binding Proteins (IGFBPs)” pathways with the contribution of upregulated *Igfb3* and downregulated *Penk* and *Spp1* genes was found after the exposure to low and medium doses of CuO and ZnO NPs. IGFs are paracrine factors that have a major role in MSC-mediated wound healing. They enhance proliferation and promote pluripotency/self-renewal of MSCs. IGFs are also implicated in MSC osteogenic and adipogenic differentiation [71]. PENK is the precursor of the endogenous opioid enkephalin, which is involved in the regulation of stem cell proliferation and stress response [72], while SPP1 controls cell adhesion, survival, migration, and immune regulation and serves as a major marker of osteogenesis. The inhibition of *Spp1* expression may lead to promoted adipogenic differentiation and suppression of osteogenic differentiation in mice [63]. It has been documented that functional impairment of IGF may alter the MSC fate between osteogenic and adipogenic lineages [73]. IGF-binding proteins (IGFBPs) bind circulating IGFs, determine their bioavailability, and modify their activity. Several studies have demonstrated that modulation of the IGFBP3 level led to changes in MSC differentiation [74,75]. In relation to our study, deregulation of *Igfb3* and other IGF-related proteins may further contribute to the positive effect of ZnO and CuO NP adipogenesis.

In contrast to Ag, CuO and ZnO NPs are not routinely used in medical practice, and their effect has been less studied. The effect of ZnO NPs together with MSCs was recently investigated in Norozi et al. (2024) [33]. A polyurethan scaffold with the addition of ZnO significantly supported the growth of MSCs and enhanced osteogenic differentiation. Promoted osteogenic differentiation was also observed in other studies [32,76]. Enhanced osteogenesis was observed in CuO- and ZnO-exposed MSCs, while adipogenesis was rather reduced by CuO, and ZnO exhibited no effect [35]. No effect of CuO was found in another study [34]. Overall, similarly to Ag NPs, no consensual effect of ZnO and CuO NPs on MSC differentiation has been demonstrated.

### 3.8. miRNA Profiling

Recent studies have demonstrated that miRNAs are critical regulators of MSC differentiation, paracrine activity, and other cellular processes such as proliferation, survival, and migration. In our study, we identified a set of miRNAs which were significantly differentially expressed for each condition and subjected them to a linear regression analysis to find significant interactions with target DEGs. We further performed miRNA-target enrichment analysis with the miRNAs significantly differentially expressed after the exposure to a high dose of Ag NPs and a low dose of CuO NPs, as these conditions only generated a sufficient number of DEmiRNAs necessary for such analysis. MiR-126 (Mmu-mir-126-3p) exhibited the highest number of significant interactions with DEGs following exposure to all NPs in our study. Although no verified interaction identified in public interaction databases was found, MiR126 contributed to the overrepresentation of most KEGG pathways that were selected by the miRNA-target enrichment analysis, suggesting its importance in numerous processes. MiR-126 has been correlated with immune- and inflammation-related diseases, such as diabetic mellitus, chronic obstructive pulmonary disease, rheumatoid arthritis, as well as cancer [77,78,79]. It has been demonstrated that overexpressed MiR-126 promotes cell proliferation, migration, invasion, and endothelial differentiation while inhibiting cell apoptosis and osteogenic differentiation in human-bone-derived MSCs via upregulation of PI3K/AKT and MEK1/ERK1 signaling pathways [80]. Similarly, the enhanced expression of MiR-126 was also found in our study. We further observed modulation of various KEGG pathways (MAPK signaling pathway, proteoglycans in cancer, and thyroid hormone signaling pathway) following exposure to a high dose of Ag NPs and a low dose of CuO NPs with the contribution of MiR-126. The most frequently represented targets of MiR-126 in these processes (*Akt1, Akt2, Kras, Pik3gc, Pik3r1,* and *Pik3r2*) indicate that PI3K/AKT signaling is the key pathway affected by MiR-126 and may possibly affect multiple cellular processes including differentiation of MSCs, resulting in a positive effect on wound healing.

MiR-92a (mmu-MiR-92a-1-5p) was specifically downregulated after the exposure to ZnO NPs only. Although the number of interactions with DEGs was revealed in our study, none of these interactions is evidenced in interaction databases. Previously it was described that this miRNA was expressed in mouse embryonic stem cells as in adult tissues and probably belongs to a group of miRNAs that regulate the general aspects of cell physiology [81]. It has been reported that upregulated MiR-92a in human MSCs suppresses the angiogenic properties of these cells that are important for tissue regeneration [82]. MiR-92a has also been shown to inhibit adipogenesis of MSCs and thus complicate the regeneration during anti-cancer therapy [83]. On the contrary, our study revealed that expression of MiR-92a was suppressed. This may in turn enhance the healing potential of MSCs.

## 4. Materials and Methods

### 4.1. Cell Culture

In all experiments, female BALB/c mice at the age of 10–16 weeks were used. The animals were obtained from the Institute of Molecular Genetics of the Czech Academy of Sciences in Prague. The use of animals was approved by the local Animal Ethics Committee of the Institute of Experimental Medicine of the Czech Academy of Science in Prague.

MSCs were isolated from inguinal fat pads. The small pieces of tissue were digested for 60 min in 37 °C with a 1% solution of collagenase I (Sigma-Aldrich, Saint Louis, MO, USA) in Hanks’ balanced salt solution with Ca^2+^ and Mg^2+^. The cell suspension was washed and seeded in 15 mL of Dulbecco’s modified Eagle’s medium (DMEM, Sigma-Aldrich) containing 10% fetal bovine serum (FBS, Gibco BRL, Grand Island, NY, USA), antibiotics (100 U/mL of penicillin, 100 µg/mL of streptomycin), and 10 mM HEPES buffer (referred to as complete DMEM) in 75 cm^2^ tissue culture flasks (Techno Plastic Products, TPP, Trasadingen, Switzerland). After 48 h of cultivation, the non-adherent cells were washed out, and the adherent cells were cultured for 14 days with a regular exchange of the medium and passaging of the cells to maintain their optimal concentration. In all the experiments, the cells harvested in the 3rd to 4th passage were used.

MSCs were adherent to plastic surfaces and had a typical fibroblast-like morphology. Phenotypic characterization of MSCs was performed by using an LSRII flow cytometer (BD Bioscience, Franklin Lakes, NJ, USA) and analyzed using FlowJo 10 software (Tree Star, Ashland, OR, USA) as previously described in detail [23,35,36]. In brief, MSCs were stained with anti-mouse monoclonal antibodies: allophycocyanin (APC)-labeled anti-CD44 (clone IM7, BD PharMingen, San Jose, CA, USA), phycoerythrin (PE)-labeled anti-CD105 (clone TY/11.8, eBioscience, San Diego, CA, USA), fluorescein isothicyanate (FITC)-labeled anti-CD90.2 (clone 30-H12, BioLegend, San Diego, CA, USA), FITC-labeled anti-CD45 (clone 30-F11, BioLegend), APC-labeled anti-CD11b (clone M1/70, BioLegend), and PE-labeled anti-CD31 (clone MEC 13.3, BD PharMingen) in PBS for 30 min at 4 °C. Cells stained with PE-labeled rat IgG2a (clone RTK2758, BioLegend), APC-labeled rat IgG2b (clone RTK4530, BioLegend), and FITC-labeled rat igG2b (clone eB149/10H5, eBioscience) were used as negative controls. Dead cells were stained using Hoechst 33,258 fluorescent dye (Invitrogen, Carlsbad, CA, USA), added to the samples 10 min before the flow cytometry analysis.

### 4.2. Nanomaterials and Exposure Design

Three different metal nanomaterials were applied to MSCs in this study—Ag and CuO (Sigma-Aldrich) and ZnO (JRC Nanomaterials Repository). Their physico-chemical properties have been described previously [23,35,36]. These results are summarized in Appendix A. For the transcriptomic analyses, three non-cytotoxic concentrations of each nanomaterial were selected based on previous results of cytotoxicity tests [36]: Ag NPs (low—1.5; medium—3.125; high—6.25 µg/mL), CuO (low—0.2; medium—0.3; high—0.4 µg/mL), and ZnO (low—0.75; medium—1.5; high—3.125 µg/mL). MSCs were exposed to NPs for 4, 24, and 48 h. The data for individual time periods were used as covariates in statistical analyses and thus are not reported separately. All tested conditions were carried out in biological duplicate in three independent experiments (hexaplicates in total).

### 4.3. RNA Isolation, Library Preparation, and Sequencing

The total RNA was isolated from the harvested cells (hexaplicates, as reported above) with the AllPrep DNA/RNA Mini Kit (Qiagen, Hilden, Germany) according to the protocol. The concentration of extracted RNA was measured with the HS RNA kit by the Qubit 4 fluorometer (both Thermo Fisher Scientific, Wilmington, DE, USA), and the RNA integrity number was checked with the SS RNA kit by the Fragment Analyzer (both Agilent Technologies, Santa Clara, CA, USA).

The total RNA (200 ng) was used for mRNA selection with NEBNext Poly (A) mRNA Magnetic Isolation Module; mRNA libraries were prepared with NEBNext Ultra II Directional RNA Library Prep with Beads and NEBNext Multiplex Oligos for Illumina (all New England Biolabs, Ipswitch, MA, USA). The total RNA (100 ng) was used for the miRNA libraries preparation with the QIAseq miRNA Library Kit and QIAseq miRNA 96 Index IL (both Qiagen). The mRNA and miRNA libraries were assembled based on the manufacturer’s instructions. The concentration of both types of libraries was checked with the 1× dsDNA HS kit (Thermo Fisher Scientific) on the Qubit 4 fluorometer, and their profile and size were analyzed by the Fragment Analyzer with the HS NGS Fragment kit (Agilent Technologies).

For mRNA libraries, the pair-end (2 × 60 cycles) sequencing was used, and for miRNA libraries, the single-end (85 cycles) sequencing with the NovaSeq 6000 Reagent Kit v1.5 (100 cycles) on the NovaSeq 6000 instrument was used (all Illumina, USA). As the samples were sequenced in separate lanes, the reads in FASTQ format resulting from the same samples were concatenated. miRNA primary analysis of the FASTQ files was performed using Qiagen GeneGlobe Analysis Center, which included UMI-based read deduplication and mapping to miRbase.

### 4.4. Data Analysis

We utilized the nf-core pipelines for standardized and reproducible RNA-seq data analysis [84]. These pipelines integrate various bioinformatics tools for quality control, alignment, and quantification, ensuring a consistent workflow with robust results. Specifically, we employed the nf-core/rnaseq pipeline version 3.2 with the GRCm38 reference genome for mRNA FASTQ processing and the nf-core/smrnaseq pipeline version 2.2.4 with the same reference genome and the Nextflex protocol was used for miRNA quantification.

The differential expression analysis was performed in the R environment with the DESeq2 library [85]. The library was used to normalize the read counts and to identify differences in gene expression between treatment groups. The treatment groups were defined by the exposure material (clean culture medium, Ag, ZnO, and CuO) and its dose; the exposition length served as a model covariate.

The mRNA–miRNA correlation analysis was conducted in the R environment. First, sets of differentially expressed mRNAs and miRNAs were identified for each nanoparticle. The detection thresholds were set to an adjusted *p*-value < 0.01 and |log_2_FC| > 0.58, with DESeq as the detection method. For an mRNA or miRNA to be considered differentially expressed, it had to meet the criteria for at least one of the three tested doses.

Correlations between all candidate mRNA and miRNA pairs were calculated independently for each of the three nanoparticles, with the doses merged (resulting in 24 samples for each nanoparticle). The dose was included as an explanatory variable. Pairs with significantly adjusted correlation *p*-values were identified and used to construct mRNA–miRNA interaction networks.

The interaction networks were further expanded using validated and predicted miRNA target information obtained from the multiMiR package [86], which integrates data from 14 human and mouse miRNA target databases. The predicted and validated targets were used to highlight edges in the mRNA–miRNA interaction correlation networks and to identify the miRNAs with the highest number of differentially expressed targets.

The Venn diagrams for presenting common or unique gene deregulation or biological pathway affection were created using the Bioinformatics & Evolutionary Genomics tool (http://bioinformatics.psb.ugent.be/webtools/Venn/, accessed on 9 July 2025). The biological pathways affected by mRNA and miRNA deregulation were analyzed by DAVID [87,88] and miRPath v.3 [89], respectively.

## 5. Conclusions

In the current study, we revealed a number of biological processes and pathways modulated in response to the exposure of MSCs with antimicrobial metal-based NPs. The results strongly suggest that all NPs possessed an immunomodulatory effect on MSCs, including the generation of extracellular vesicles, which represent an alternative to cell-based therapy. Moreover, deregulation of IFN-stimulated genes mediating an immunosuppressive effect was observed following the exposure to Ag NPs. These processes could be considered as beneficial as they can enhance the therapeutic efficacy of MSCs. Other processes such as HDL remodeling deregulated by Ag NPs and the regulation of Insulin-like Growth Factor transport and uptake affected by CuO and ZnO NPs rather imply that NPs interfere with the differentiation of MSCs, suggesting suppression of osteogenesis and a shift towards adipogenesis. The deregulation of genes involved in calcium signaling in response to all NPs and the organization of the extracellular matrix in response to ZnO NPs and Ag NPs further indicate the initiation of adipogenesis. We also revealed upregulation of ECM-related genes that has been linked to tissue regeneration. Recent studies suggest that adipocytes and their progenitors substantially contribute to tissue repair and regeneration. However, since the control of the lineage-specific differentiation is necessary for various MSC therapy applications, the use of the tested NPs is problematic as studies investigating how NPs influence the differentiation process provide inconsistent conclusions. Importantly, Ag NPs have been shown to increase the expression of antioxidant genes as a response to the excessive production of ROS. This may further limit the use of Ag NPs.

miRNA profiling revealed upregulation of MiR-126 by all NPs. MiR-126 exhibited the highest number of interactions with mRNA targets. Enhanced expression of this miRNA may lead to modulation of multiple cellular processes including enhanced differentiation in MSCs that may potentially improve wound healing. Another highly interacting miRNA, MiR-92, was downregulated by ZnO NPs only. This effect might be related to the enhancement of processes such as adipogenesis and angiogenesis that contribute to healing and tissue regeneration.

Our study has a limitation that needs to be acknowledged. Since functional experiments were not performed, the specific mechanisms proposed in our study have not been validated. We focused on screening of the potentially modulated genes and processes at the transcriptional level as well as the involvement of modulated miRNAs expression; however, the functional roles and mechanisms of these genes and miRNAs expression changes should be clarified and validated in the future.

## Figures and Tables

**Figure 1 ijms-26-07583-f001:**
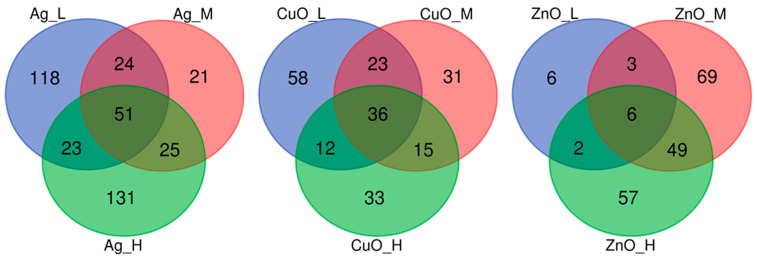
The distribution of common and unique DEGs after exposure of MSCs to three different doses (L—low; M—medium; H—high) of three nanomaterials (Ag, CuO, and ZnO NPs).

**Figure 2 ijms-26-07583-f002:**
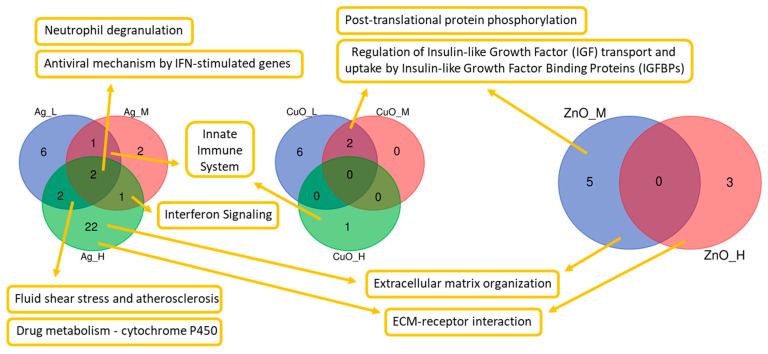
Numbers of affected biological pathways after exposure of MSCs to three different doses (L—low; M—medium; H—high) of three nanomaterials (Ag, CuO, and ZnO NPs). Ten pathways found in more than one comparison are shown by name.

**Figure 3 ijms-26-07583-f003:**
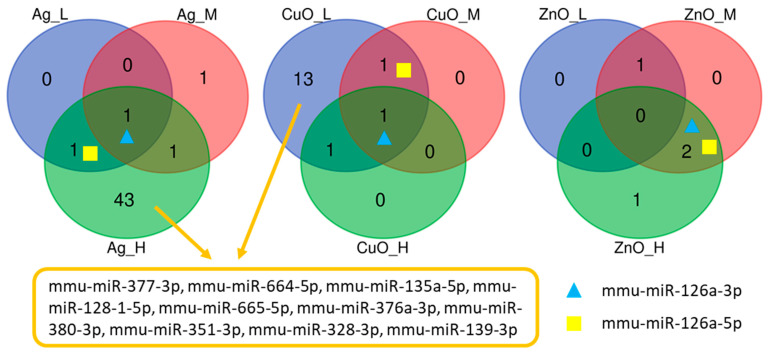
The distribution of common and unique DEmiRNAs after exposure of MSCs to three different doses (L—low; M—medium; H—high) of nanomaterials (Ag, CuO, and ZnO NPs).

**Figure 4 ijms-26-07583-f004:**
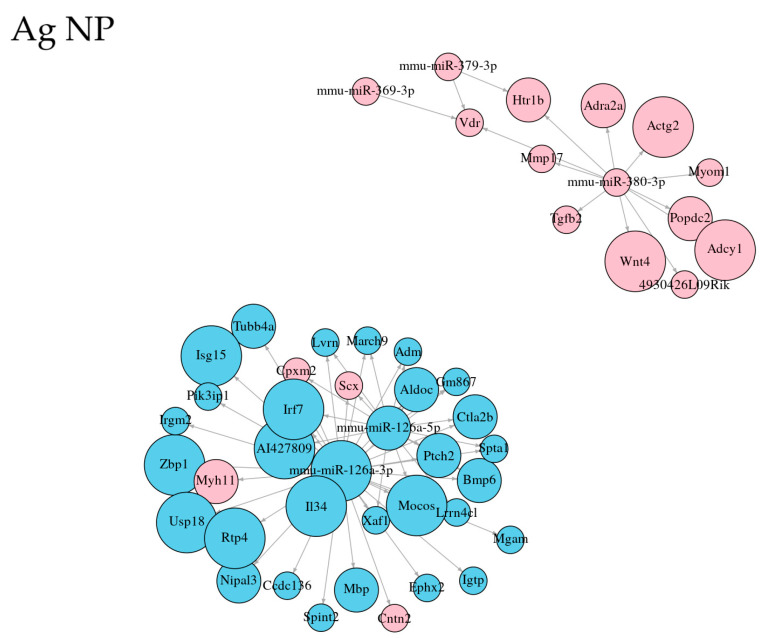
Plots showing the top 50 significant interactions of DEmiRNAs-DEGs after the exposure of MSCs to three nanomaterials (Ag, CuO, and ZnO NPs). Blue and red color of the nodes indicates RNA upregulation and downregulation, respectively. The size of the nodes represents the strength of deregulation (log_2_fold change).

**Table 1 ijms-26-07583-t001:** Number of downregulated, upregulated, and the total number of DEGs after the exposure of MSCs to three nanomaterials in three different doses.

NP	Dose	Down	Up	Total
Ag	Low	80	136	216
	Medium	41	80	121
	High	81	149	230
CuO	Low	66	63	129
	Medium	61	44	105
	High	55	41	96
ZnO	Low	13	4	17
	Medium	43	84	127
	High	58	56	114

**Table 2 ijms-26-07583-t002:** Top 5 commonly down- and upregulated genes after exposure of MSCs to three different doses of three nanomaterials (Ag, CuO, and ZnO NPs).

NP	Ensembl ID	Gene	Log_2_FC	Adjusted *p*-Value
Ag	ENSMUSG00000028655	*Mfsd2a*	−1.36	5.67 × 10^−9^
	ENSMUSG00000060284	*Sp7*	−1.14	1.29 × 10^−10^
	ENSMUSG00000036264	*Fstl4*	−1.13	5.72 × 10^−4^
	ENSMUSG00000020542	*Myocd*	−1.10	1.01 × 10^−2^
	ENSMUSG00000028766	*Alpl*	−1.04	2.04 × 10^−10^
	ENSMUSG00000091971	*Hspa1a*	1.88	8.67 × 10^−11^
	ENSMUSG00000031765	*Mt1*	2.02	1.17 × 10^−12^
	ENSMUSG00000064247	*Plcxd1*	2.06	6.63 × 10^−6^
	ENSMUSG00000031762	*Mt2*	2.32	8.69 × 10^−12^
	ENSMUSG00000022602	*Arc*	2.77	3.65 × 10^−15^
CuO	ENSMUSG00000038642	*Ctss*	−1.47	2.10 × 10^−7^
	ENSMUSG00000025044	*Msr1*	−1.35	1.81 × 10^−2^
	ENSMUSG00000060284	*Sp7*	−1.35	1.66 × 10^−15^
	ENSMUSG00000046805	*Mpeg1*	−1.30	1.50 × 10^−5^
	ENSMUSG00000003283	*Hck*	−1.14	4.06 × 10^−4^
	ENSMUSG00000033377	*Palmd*	0.88	2.59 × 10^−4^
	ENSMUSG00000031871	*Cdh5*	0.90	2.71 × 10^−3^
	ENSMUSG00000020826	*Nos2*	0.93	3.30 × 10^−6^
	ENSMUSG00000031765	*Mt1*	1.16	2.52 × 10^−5^
	ENSMUSG00000031762	*Mt2*	1.30	5.34 × 10^−5^
ZnO	ENSMUSG00000046167	*Gldn*	−1.27	1.29 × 10^−6^
	ENSMUSG00000060284	*Sp7*	−1.14	2.81 × 10^−12^
	ENSMUSG00000004371	*Il11*	−1.09	2.24 × 10^−11^
	ENSMUSG00000028766	*Alpl*	−0.97	1.65 × 10^−31^
	ENSMUSG00000046352	*Gjb2*	−0.67	4.62 × 10^−5^
	ENSMUSG00000037362	*Nov*	0.97	3.95 × 10^−8^

The DEGs for this table were chosen based on the lowest/highest log_2_ fold change after exposure to the high dose of Ag, CuO, and ZnO NPs. Log_2_ fold change (Log_2_FC) and adjusted *p*-values are indicated for individual genes.

**Table 3 ijms-26-07583-t003:** List of the top affected mRNA pathways after exposure to the tested NPs.

NP	Scheme	Pathway	Genes#	*p*-Value	Genes	
Ag_L	REACTOME	Neutrophil degranulation	14	4.6 × 10^−3^	↑	*Anpep*, *Aldoc*, *Gca*, *Hspa1a*, *Hspa1b*, *H2-Q2*, *Hvcn1*, *Ifi205*, *Mndal*, *Pecam1*, *Syngr1*, *Trpm2*
					↓	*Mmp9*, *Slpi*
	REACTOME	HDL remodeling	3	5.3 × 10^−3^	↑	*Abcg1*, *Apoe*, *Pltp*
	REACTOME	Innate Immune System	19	1.4 × 10^−2^	↑	*Anpep*, *Aldoc*, *C1s2*, *Gca*, *Hspa1a*, *Hspa1b*, *H2-Q2*, *Hvcn1*, *Ifi205*, *Irf7*, *Mndal*, *Pecam1*, *Syngr1*, *Trpm2*, *Usp18*
					↓	*Hck*, *Mmp9*, *Slpi*, *Tec*
Ag_M	REACTOME	Antiviral mechanism by IFN-stimulated genes	4	1.7 × 10^−2^	↑	*Isg15*, *Hspa1a*, *Hspa1b*, *Usp18*
	REACTOME	Neutrophil degranulation	8	3.3 × 10^−2^	↑	*Hspa1a*, *Hspa1b*, *H2-Q2*, *Hvcn1*, *Mgam*, *Pecam1*, *Syngr1*
					↓	*Mmp9*
	WIKI	Calcium regulation in cardiac cells	5	3.4 × 10^−2^	↑	*Rgs2*
					↓	*Adcy1*, *Gjb2*, *Gjb3*, *Rgs4*
Ag_H	KEGG	Drug metabolism–cytochrome P450	10	1.8 × 10^−7^	↑	*Aox1*, *Fmo1*, *Fmo2*, *Gsta1*, *Gsta3*, *Gstm1*, *Gstm6*, *Gstp2*, *Mgst1*, *Mgst2*
	REACTOME	Glutathione conjugation	7	7.3 × 10^−6^	↑	*Gsta1*, *Gsta3*, *Gstm1*, *Gstm6*, *Gstp2*, *Mgst1*, *Mgst2*
	KEGG	Fluid shear stress and atherosclerosis	11	1.1 × 10^−5^	↑	*Nqo1*, *Cdh5*, *Gsta1*, *Gsta3*, *Gstm1*, *Gstm6*, *Gstp2*, *Hmox1*, *Mgst1*, *Mgst2*
					↓	*Mmp9*
CuO_L	REACTOME	Post-translational protein phosphorylation	7 *	2.5 × 10^−4^	↑	*Apoe*, *Igfbp3*, *Scg2*, *Trf*
	REACTOME	Regulation of Insulin-like Growth Factor (IGF) transport and uptake by Insulin-like Growth Factor Binding Proteins (IGFBPs)		3.3 × 10^−4^	↓	*Notum*, *Penk*, *Spp1*
	KEGG	Calcium signaling pathway	7	5.4 × 10^−3^	↑	*Ednrb*, *Fgf9*, *Gna14*, *Mst1r*, *Nos2*
					↓	*Adcy1*, *Mylk2*
CuO_M	REACTOME	Post-translational protein phosphorylation	4 *	3.5 × 10^−2^	↑	*Vwa1*
	REACTOME	Regulation of Insulin-like Growth Factor (IGF) transport and uptake by Insulin-like Growth Factor Binding Proteins (IGFBPs)		4.0 × 10^−2^	↓	*Notum*, *Penk*, *Spp1*
CuO_H	REACTOME	Innate Immune System	11	2.8 × 10^−2^	↑	*Atp6v0a4*, *Hvcn1*, *Myh2*, *Nos2*
					↓	*Atp8a1*, *Adam8*, *Ctss*, *Hck*, *Lyz2*, *Mmp9*, *Slc11a1*
ZnO_M	REACTOME	Post-translational protein phosphorylation	7 *	1.1 × 10^−4^	↑	*Cp*, *Igfbp3*, *Scg2*, *Vwa1*
	REACTOME	Regulation of Insulin-like Growth Factor (IGF) transport and uptake by Insulin-like Growth Factor Binding Proteins (IGFBPs)		1.5 × 10^−4^	↓	*Notum*, *Penk*, *Spp1*
	REACTOME	Extracellular matrix organization	8	1.2 × 10^−3^	↑	*Col28a1*, *Eln*, *Fbn2*, *Lum*, *Mfap4*, *Pecam1*
					↓	*Spp1*, *Tnn*
ZnO_H	KEGG	cGMP-PKG signaling pathway	6	3.0 × 10^−3^	↑	*Ednrb*, *Rgs2*
					↓	*Adcy1*, *Adra2a*, *Nppb*, *Prkg2*
	KEGG	ECM-receptor interaction	4	1.5 × 10^−2^	↑	*Thbs4*
					↓	*Npnt*, *Spp1*, *Tnn*
	KEGG	Relaxin signaling pathway	4	3.9 × 10^−2^	↑	*Ednrb*, *Nos2*
					↓	*Adcy1*, *Mmp9*

DEGs involved in pathways are indicated with their deregulation direction (↑ signs upregulation and ↓ signs downregulation). The symbol **#** means number. Asterisks (*) indicate the same genes involved in two pathways [Post-translational protein phosphorylation and Regulation of Insulin-like Growth Factor (IGF) transport and uptake by Insulin-like Growth Factor Binding Proteins (IGFBPs)]. Letters _L, _M, and _H denote low, medium, and high concentration of the respective NP.

**Table 4 ijms-26-07583-t004:** Number of downregulated, upregulated, and the total number of DEmiRNAs after the exposure of MSCs to three nanomaterials in three different doses.

NP	Concentration	Down	Up	Total
Ag	Low	0	2	2
	Medium	1	2	3
	High	20	26	46
CuO	Low	7	9	16
	Medium	0	2	2
	High	1	1	2
ZnO	Low	0	1	1
	Medium	0	3	3
	High	1	2	3

**Table 5 ijms-26-07583-t005:** The top 5 most significant KEGG pathways affected by miRNA deregulation after exposure to a high dose of Ag and a low dose of CuO NPs.

NP	KEGG Pathway	ID	*p*-Value	Genes#	miRNAs#
Ag High	Proteoglycans in cancer	mmu05205	2.47 × 10^−5^	8	34
	Biosynthesis of unsaturated fatty acids	mmu01040	4.77 × 10^−5^	11	16
	Hippo signaling pathway	mmu04390	4.77 × 10^−5^	60	33
	Thyroid hormone signaling pathway	mmu04919	7.45 × 10^−5^	53	34
	MAPK signaling pathway	mmu04010	8.17 × 10^−5^	104	36
CuO Low	Phosphatidylinositol signaling system	mmu04070	2.25 × 10^−4^	8	9
	Thyroid hormone signaling pathway	mmu04919	2.25 × 10^−4^	25	11
	Axon guidance	mmu04360	2.25 × 10^−4^	31	12
	Proteoglycans in cancer	mmu05205	3.19 × 10^−4^	41	11
	Signaling pathways regulating pluripotency of stem cells	mmu04550	1.03 × 10^−3^	29	11

The *p*-values, number of DEmiRNAs, and affected genes are indicated for each KEGG pathway. The symbol **#** means number.

**Table 6 ijms-26-07583-t006:** An overview of miRNA–mRNA interactions.

NP	Number of miRNAs	miRNA	Number of Interactions
Ag	5	mmu-miR-126a-3p	33
		mmu-miR-126a-5p	4
		mmu-miR-369-3p	1
		mmu-miR-379-3p	1
		mmu-miR-380-3p	11
CuO	3	mmu-miR-126a-3p	22
		mmu-miR-126a-5p	19
		mmu-miR-142a-3p	2
ZnO	4	mmu-miR-126a-3p	3
		mmu-miR-126a-5p	2
		mmu-miR-467d-3p	3
		mmu-miR-92a-1-5p	42

## Data Availability

The data presented in this study are available on request from Pavel Rössner (rossner.pavel@iem.cas.cz).

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
