# Peer review of "Transcriptomic Profiling of Mouse Mesenchymal Stem Cells Exposed to Metal-Based Nanoparticles"

_ijms, 2025, doi:10.3390/ijms26157583_

Round 1

Reviewer 1 Report

Comments and Suggestions for Authors

Despite 230 DEGs for Ag NPs (high dose), only 27 pathways were enriched (Table 3). Key processes like "adipogenesis enhancement" rely on indirect markers (Apoe, Ptlp) without adipogenic pathway enrichment. 

Venn diagrams lack clear labels for overlapping/non-overlapping regions (e.g., "118" in Ag_L section). Total unique DEGs per NP are not quantified. 

Background details for the transcriptomic assay can be enhanced by referencing 10.1021/acsnano.5c07535; 10.1016/j.celbio.2025.100020; additionally, what strengths does the present study offer compared to published articles in related areas?

Pathway names (e.g., "ECM+ receptor interaction") contain typographical errors ("+" likely unintended).

Interaction plots (p. 11) are described but omitted, preventing assessment of miRNA-mRNA network topology (e.g., hub miRNAs). 

Results suggest NPs universally suppress osteogenesis (via Sp7 downregulation), yet cited literature reports NP-enhanced osteogenesis. This discrepancy is inadequately discussed. 

Author Response

Despite 230 DEGs for Ag NPs (high dose), only 27 pathways were enriched (Table 3). Key processes like "adipogenesis enhancement" rely on indirect markers (Apoe, Ptlp) without adipogenic pathway enrichment. 

Response: The detection of biological pathways affected by deregulated mRNAs was performed with DAVID software (https://david.ncifcrf.gov/) by standardized methodology. The list of deregulated mRNAs was uploaded to DAVID and all detected significant pathways (p-value < 0.05) were extracted and described in the manuscript (part 2.2, Figure 2, Supplementary Table 4). The lower number of altered pathways is probably due to variability of deregulated mRNAs. The differentiation of mesenchymal stem cells is a complex process involving multiple KEGG pathways including those related to cell cycle, apoptosis, and various metabolic processes such as ECM remodeling or lipid and glucose metabolism. Specific pathways like Wnt, TGF-beta, MAPK, PI3K-Akt signaling are known to be involved in regulating MSC differentiation. No specific pathway named “adipogenesis” or “osteogenesis” is defined by KEGG pathway database. Pathway analysis helps identify key genes that are differentially expressed during differentiation and are enriched in specific pathways mentioned above. 

Venn diagrams lack clear labels for overlapping/non-overlapping regions (e.g., "118" in Ag_L section). Total unique DEGs per NP are not quantified. 

Response: In all Venn diagrams, all the sets are described by name of NP and its concentration (e.g. Ag_L, CuO_M, ZnO_H), which is specified in the figure legends. When the color of the section is blue, red, or green, it is related to non-overlapping genes/pathways. When the section overlaps, the number of genes/pathways in the overlapped section indicates genes/pathways deregulated/altered in more conditions. This type of visualization of common and unique deregulated genes is standardly used in these types of manuscripts and we believe, there is no need for improving it. The total unique DEGs number per NP is the sum of all numbers in each Venn diagram (for mRNA – Figure 1, for miRNA – Figure 3). We did not use this number anywhere in the text, so we believe, that our visualization of DEGs number in both these figures (from which the total number of unique DEGs can be easily calculated) is sufficient.

Background details for the transcriptomic assay can be enhanced by referencing 10.1021/acsnano.5c07535; 10.1016/j.celbio.2025.100020; additionally, what strengths does the present study offer compared to published articles in related areas?

Response: We added suggested references into Introduction, please see the revised version, line 75. Regarding the innovative aspects of the current study, we used whole-genome gene expression profiling in order to identify all the processes and pathways that are potentially altered in MSCs in response to the exposure to NPs (including those that have not been revealed before) and may thus modify the healing potency of MSCs. We also focused on epigenetic mechanism by which the transcription of specific target genes may be regulated. This is described in Introduction, last paragraph. To our knowledge, no study provided such broad screening of potential effects that are caused by the action of NPs. Studies reporting effects of NPs on individual processes in MSCs, e.g. differentiation or ECM organization are included in the Discussion. Moreover, while relatively large number of studies describe effects of Ag NPs in MSCs, much less studies are concerned with other nanomaterials such as ZnO and CuO.

Pathway names (e.g., "ECM+ receptor interaction") contain typographical errors ("+" likely unintended).

Response:  We checked the manuscript and, in all cases, where “ECM-receptor interaction” pathway is written (Table 3, Discussion – “Extracellular matrix organization, ECM-receptor interaction”), it is typed correctly as “ECM-receptor interaction”, we did not find and “+” related to this pathway.

Interaction plots (p. 11) are described but omitted, preventing assessment of miRNA-mRNA network topology (e.g., hub miRNAs). 

Response: In the Results, there is a whole section dedicated to miRNA-mRNA interactions (2.5.). This section includes one figure (Figure 4) with the interaction network created from 50 most significant interactions, one table (Table 6) with summary of the most interacted miRNAs, and one supplementary table (Supplementary Table 7) with description of all detected miRNA-mRNA interactions (not only the 50 most significant). We believe that this broad set of data sufficiently describes the miRNA-mRNA analysis.

Results suggest NPs universally suppress osteogenesis (via Sp7 downregulation), yet cited literature reports NP-enhanced osteogenesis. This discrepancy is inadequately discussed. 

Response:  In the Discussion, we presented several studies that report the contradictory effect of Ag NPs on differentiation of MSCs. While some of them described enhanced osteogenesis (references no. 51, 52), others reported inhibited osteogenesis and adipogenesis (23), enhanced adipogenesis in comparison to osteogenesis (53), enhanced chondrogenesis (26) or observed no effect (54). Although several studies showed rather positive effect on osteogenesis, we also presented the one which reports an opposite effect and is in accordance to our results. As a reason of this discrepancy, we presented possible factors such as concentration, surface coating or size (lines 445 and 446). We added references, please see the revised version.

Reviewer 2 Report

Comments and Suggestions for Authors

In this work, the authors investigated the effect of combining mesenchymal stem cells with nanoparticles as a treatment for tissue injuries by analysing changes in mRNA and miRNA expression, deregulated processes, and mRNA-miRNA interactions, after in vitro exposure to nanoparticles. The manuscript is clear and detailed, but the results are not well described since a reader who is not familiar with the experimental techniques used in this work cannot understand the achieved results or the conclusions; therefore, it can be considered for publication, if the following major revision is considered:

  • A diagram showing and explaining the different steps taken to obtain the results should be added to the manuscript to understand the data listed.
  • References should be included in the materials and methods section.

Author Response

In this work, the authors investigated the effect of combining mesenchymal stem cells with nanoparticles as a treatment for tissue injuries by analysing changes in mRNA and miRNA expression, deregulated processes, and mRNA-miRNA interactions, after in vitro exposure to nanoparticles. The manuscript is clear and detailed, but the results are not well described since a reader who is not familiar with the experimental techniques used in this work cannot understand the achieved results or the conclusions; therefore, it can be considered for publication, if the following major revision is considered:

A diagram showing and explaining the different steps taken to obtain the results should be added to the manuscript to understand the data listed.

Response: We believe that the pipeline of this project was straightforward and standard for this type of manuscript, so it does not necessarily need any diagram showing the whole procedure. We summarized the process in the graphical abstract which we unfortunately forgot to include with the original submission. The graphical abstract was submitted together with the revisions.

References should be included in the materials and methods section.

Response: The Material and Methods section is divided into 4 sub-sections and among three of them, references are included. Specifically, 4.1. and 4.2 include 3 references [21,33,34] about previously published methods of working with the mesenchymal stem cells and about determining the physico-chemical properties and non-cytotoxic concentrations of NPs. Subsection 4.4. includes 6 references [81-86] regarding the data analysis and used software. The only sub-section without the references is 4.3. (RNA isolation, library preparation, sequencing) where we refer to manufacturer’s instructions.

All these 9 references are included in the Reference list at the end of the manuscript.

We suppose that this number of references is sufficient for this section.

Reviewer 3 Report

Comments and Suggestions for Authors

The manuscript, entitled "Transcriptomic profiling of mouse mesenchymal stem cells exposed to metal-based nanoparticles", publishes the results of a study on the effect of metal nanoparticles on mesenchymal stem cells. The authors carried out a comprehensive work and analyzed the results obtained. The work is of scientific interest, as the issue of the use of metal nanoparticles in medicine is controversial and debatable. The data from different authors are contradictory, so additional research may clarify the situation. The text is written correctly, but it is not always clear, and an explanation of the authors' thoughts is required. It is advisable to facilitate the presentation of the material by referring to the main topic of the work. The conclusions are supported by the results, however, the abstract is written too promisingly. The graphic design is satisfactory, and the number of references to literary sources is sufficient. It is recommended to accept the article after minor changes.

1)Formal remarks on the design of the article. There is no caption to Table 3. Figure 3 and its caption are on different pages. It is necessary to number the sub-chapters of Chapter 3. The font of the text in Figure 4 differs from the font of the main part of the article. There is too much free space at the end of page 6. Table 1 is divided into two parts located on different pages.

2) Based on the description of the methods and results, the work is essentially a study of the toxicity of metal nanoparticles for MSCs. As for the joint healing effect of MSC and NP, which was stated in the summary, this topic was not disclosed.

3) Given that metal nanoparticles are characterized by cytotoxicity, it is possible that the intended healing effect may be due to the fact that reactive oxygen species and other negative factors of nanoparticles cause an immune response at the body level, which also contributes to accelerated healing. Differentiation of MSCs can also take place as a response to a negative action. In other words, nanoparticles have not so much a therapeutic effect as they stimulate the immune system. This idea could be revealed in the discussion.

4) The widespread mention by the authors of violations of RNA regulation should be explained in more understandable words, indicating how these violations are related or may be related to the healing of damaged tissues. Gene expression can occur in response to contact with nanoparticles, and the fact that the authors identify expressed genes that are independent of dose was mentioned in passing [133-139] and not properly explained. 

Author Response

The manuscript, entitled "Transcriptomic profiling of mouse mesenchymal stem cells exposed to metal-based nanoparticles", publishes the results of a study on the effect of metal nanoparticles on mesenchymal stem cells. The authors carried out a comprehensive work and analyzed the results obtained. The work is of scientific interest, as the issue of the use of metal nanoparticles in medicine is controversial and debatable. The data from different authors are contradictory, so additional research may clarify the situation. The text is written correctly, but it is not always clear, and an explanation of the authors' thoughts is required. It is advisable to facilitate the presentation of the material by referring to the main topic of the work. The conclusions are supported by the results, however, the abstract is written too promisingly. The graphic design is satisfactory, and the number of references to literary sources is sufficient. It is recommended to accept the article after minor changes.

1)Formal remarks on the design of the article. There is no caption to Table 3. Figure 3 and its caption are on different pages. It is necessary to number the sub-chapters of Chapter 3. The font of the text in Figure 4 differs from the font of the main part of the article. There is too much free space at the end of page 6. Table 1 is divided into two parts located on different pages.

Response: We adjusted the mentioned captions and fonts. We believe that most of these issues will be solved by the journal redaction during the final text formatting (e.g. dividing of the tables, too much space…). The caption of Table 3 was added and the legend of this table was corrected. The number of sub-sections of Chapter 3 (Discussion) were added. The font of the labels in Figure 4 was changed to Palatino Linotype to correspond with the rest of the manuscript.

2) Based on the description of the methods and results, the work is essentially a study of the toxicity of metal nanoparticles for MSCs. As for the joint healing effect of MSC and NP, which was stated in the summary, this topic was not disclosed.

Response: We aimed to test the toxicity and biocompatibility of metal NPs with MSCs since they are intended to be simultaneously used with MSCs as antimicrobial agents. Their primary role is supposed to prevent the bacterial infection thus supporting wound healing. In this regard, we modified a summary, please see the Abstract in the revised version.

3) Given that metal nanoparticles are characterized by cytotoxicity, it is possible that the intended healing effect may be due to the fact that reactive oxygen species and other negative factors of nanoparticles cause an immune response at the body level, which also contributes to accelerated healing. Differentiation of MSCs can also take place as a response to a negative action. In other words, nanoparticles have not so much a therapeutic effect as they stimulate the immune system. This idea could be revealed in the discussion.

Response: NPs can be used for their antibacterial and anti-inflammatory properties, but could also serve as carriers of different therapeutic agents. In addition to their therapeutic effects, NPs could act as targeting agent to deliver stem cells to injured tissue (https://doi.org/10.2147/IJN.S527928). Importantly, in our study, significantly increased production of ROS was detected in MSCs after the exposure to high dose of Ag NPs only but the immunomodulatory effect exerted all the tested NPs. Multiple mechanisms by which NPs stimulate or inhibit immune system exist. More detailed mechanism by which NPs cause immunomodulatory effect was reported in our previous study. While NPs inhibited the production of cytokines and growth factors by a direct toxic effect on MSCs in a dose-dependent manner, the preincubation of MSCs with NPs did not decrease their ability to stimulate the production of cytokines and growth factors by macrophages. It has been suggested that regulatory and therapeutic effects are also mediated by the phagocytosis of MSCs by monocytes and macrophages, and by the activation of a secretory potential of these phagocytes (DOI: 10.1007/s12015-022-10500-2). Our current data suggest production of extracellular vesicles (EV), an important immunomodulatory and therapeutical tool, whose production is potentially enhanced by the action of all the tested NPs. However, the mechanism behind (production of EV due to the increased ROS) could not be inferred from our data. Functional assays should be used to verify this mechanism. Moreover, this does not explain the generation of EV by ZnO and CuO NPs, which did not generate the excessive amount of ROS. Taken together, we would prefer not to evaluate the impact of NPs on MSCs as strictly negative since we cannot verify the exact mechanism.

4) The widespread mention by the authors of violations of RNA regulation should be explained in more understandable words, indicating how these violations are related or may be related to the healing of damaged tissues. Gene expression can occur in response to contact with nanoparticles, and the fact that the authors identify expressed genes that are independent of dose was mentioned in passing [133-139] and not properly explained. 

Response: The impacts of deregulation of mRNA expression on wound healing is described in Conclusions. In summary, NPs act as immunomodulatory agents that deregulate the expression of RNAs relevant to immune response. In addition, they may affect the expression of genes involved in MSC differentiation and extracellular matrix organization. We observed dose-response changes in mRNAs expression following exposure to Ag NPs. However, numbers of genes deregulated after the exposure to other NPs do not correlate with the increasing dose. It should be noted, that changes in gene expression are not suppose to be dose dependent as completely different processes may be triggered at different doses (e.g. in our case, the effect on differentiation, specifically enhancement of adipogenesis, is mediated by low or medium doses of all NPs only and not by the high dose).

Round 2

Reviewer 2 Report

Comments and Suggestions for Authors

This manuscript has been improved considering the suggestions and, therefore, I can recommend this work for publication